# Systematic Review and Meta-Analysis Provide no Guidance on Management of Asymptomatic Bacteriuria within the First Year after Kidney Transplantation

**DOI:** 10.3390/antibiotics13050442

**Published:** 2024-05-14

**Authors:** José Medina-Polo, Eva Falkensammer, Béla Köves, Jennifer Kranz, Zafer Tandogdu, Ana María Tapia, Tommaso Cai, Florian M. E. Wagenlehner, Laila Schneidewind, Truls Erik Bjerklund Johansen

**Affiliations:** 1Department of Urology, Hospital Universitario 12 de Octubre imas12, 28040 Madrid, Spain; 2Department of Urology, Klinikum Wels-Grieskirchen, 4710 Wels, Austria; eva.falkensammer@gmail.com; 3Department of Urology, South-Pest Hospital, 1051 Budapest, Hungary; 4Department of Urology and Pediatric Urology, University Medical Center RWTH Aachen, 52074 Aachen, Germany; jennifer.kranz@rwth-aachen.de; 5Department of Urology and Kidney Transplantation, Martin Luther University, 06120 Halle, Germany; 6Department of Urology, University College London Hospitals, London NW1 2BU, UK; drzafer@gmail.com; 7Department of Urology, Hospital Universitario Río Hortega, 47012 Valladolid, Spain; amth86@gmail.com; 8Department of Urology, Santa Chiara Regional Hospital Trento, 38122, Trento, Italy; tommaso.cai@apss.tn.it; 9Institute of Clinical Medicine, University of Oslo, 0372 Oslo, Norway; tebj@uio.no; 10Clinic for Urology, Pediatric Urology and Andrology, Justus-Liebig-University Giessen, 35392 Giessen, Germany; florian.wagenlehner@chiru.med.uni-giessen.de; 11Department of Urology, University Hospital Bern, University of Bern, 3010 Bern, Switzerland; laila.schneidewind@insel.ch; 12Urology Department, Clinic for Surgery, Inflammation and Transplantation, Oslo University Hospital, 0424 Oslo, Norway; 13Institute of Clinical Medicine, University of Aarhus, 8200 Aarhus, Denmark

**Keywords:** antibiotics, asymptomatic bacteriuria (ASB), kidney transplant, urinary tract infection, graft rejection, kidney failure, meta-analysis, systematic review

## Abstract

(1) Background: Urinary tract infections (UTIs) are among the most frequent complications in kidney transplant (KT) recipients. Asymptomatic bacteriuria (ASB) may be a risk factor for UTIs and graft rejection. We aimed to evaluate available evidence regarding the benefit of screening and treatment of ASB within the first year after KT. (2) Evidence acquisition: A systematic literature search was conducted in MEDLINE, the Cochrane Library CENTRAL and Embase. Inclusion criteria were manuscripts in English addressing the management of ASB after KT. The PICO questions concerned Patients (adults receiving a KT), Intervention (screening, diagnosis and treatment of ASB), Control (screening and no antibiotic treatment) and Outcome (UTIs, sepsis, kidney failure and death). (3) Evidence synthesis: The systematic review identified 151 studies, and 16 full-text articles were evaluated. Seven were excluded because they did not evaluate the effect of treatment of ASB. There was no evidence for a higher incidence of lower UTIs, acute pyelonephritis, graft loss, or mortality in patients not treated with antibiotics for ASB. Analysis of comparative non-randomized and observational studies did not provide supplementary evidence to guide clinical recommendations. We believe this lack of evidence is due to confounding risk factors that are not being considered in the stratification of study patients.

## 1. Introduction

Urinary tract infections (UTIs) are more common in patients with end-stage renal disease and in kidney transplant (KT) recipients than in patients without these risk factors. KT recipients are immunocompromised, which is a risk factor for UTIs. Infectious complications in KT recipients are associated with increased morbidity and mortality [1,2]. Half of KT patients develop urinary tract infections within three years after transplantation [3]. The highest UTI incidence is during the first six months after the transplantation. Infections may be related to surgical injury, urethral catheterization and ureteral stent insertion [4,5]. Infections may affect graft function and even lead to graft loss and death.

Asymptomatic bacteriuria (ASB) is reported in up to 50% of KT recipients and treatment of ASB is practiced to prevent symptomatic infections [6]. Recommendations in international guidelines are, however, not clear. Guideline developers such as the European Association of Urology (EAU) and the Infectious Diseases Society of America (IDSA) state that ASB should not be treated in KT patients beyond the short-term post-transplant period, but the duration of this short term is not clearly defined [7,8,9]. On the other hand, guidelines do recommend that ASB should be diagnosed and managed in recipients with other risk factors for developing pyelonephritis, such as indwelling devices, neurogenic bladder, or a combined transplant [1]. 

The aim of our systematic review and meta-analysis was to review current evidence to try and fill identified knowledge gaps related to a possible benefit screening and treatment of ASB during specific time periods after KT (two, six, or twelve months) as well as in recipients with specific risk factors for UTI. Our primary objective was to evaluate the effect of screening and the antibiotic treatment of ASB versus no treatment on the incidence of UTI and graft survival. Our secondary objective was to look for rates of multidrug-resistant microorganisms in KT transplant recipients with UTIs. 

## 2. Results—Evidence Acquisition

The literature search identified 151 studies on the management of ASB after KT. One hundred and twenty-four records were excluded as they did not evaluate the effect of the antibiotic treatment of ASB on the incidence of urinary tract infections in KT patients or because they were case reports. Sixteen full-text articles were assessed for eligibility, and nine studies were included [10,11,12,13,14,15,16,17,18]. All nine studies included were assessed for bias. The reasons for the exclusion of records are presented in Figure 1. 

### 2.1. Comparative Studies

Five prospective, comparative and randomized trials were included, with patient numbers ranging from 88 to 205. The characteristics of the randomized studies are summarized in Table 1. The reported incidence of urinary tract infections ranged from 6% to 24% without statistically significant differences. The incidence of acute pyelonephritis ranged from 8% to 17%. Although there were no statistical differences, the study by Coussement et al. [17] reported an incidence of ASB of 33% and 53% in patients who received and did not receive antibiotics, respectively. There were no differences in terms of graft loss or mortality between patient groups in any studies. There was a higher rate of isolation of multidrug-resistant (MDR) microorganisms in patients who received antibiotic treatment for the management of ASB. The main findings of randomized studies are summarized in Table 2.

### 2.2. Observational Studies 

The review included four observational studies. The study by Green et al. [12] is a retrospective study in which the treatment of asymptomatic bacteriuria was based on the physician’s decision in each case. This study analyzed the effect of the antibiotic treatment of ASB on the incidence of urinary tract infections, graft outcomes and microbiological parameters, and reported no statistical differences between the intervention group and the control group. The characteristics and results of observational studies are summarized in Table 3 and Table 4.

### 2.3. Meta-Analyses

#### 2.3.1. Rate of Symptomatic Urinary Tract Infections

The comparison of groups with antibiotics (case) versus no treatment (control) on the rate of symptomatic urinary tract infections is summarized in Figure 2. The analysis included nine studies with 959 KT recipients, comparing antibiotics (438 KT recipients) versus no antibiotics for ASB (521 KT recipients). The study by Green et al. [12] showed a statistically significantly lower rate of UTIs in the control group of 30% versus 54% in the treatment group (*p* = 0.03). However, there was no statistically significant difference in the overall evaluation of the rate of urinary tract infections (OR 0.74, 95% CI 0.46 to 1.190, *p* = 0.21) between groups. The Q test for heterogeneity reported Q = 13.58, *p* = 0.094, tau2 = 0.209, H2 = 1.729 and I^2^ = 42.2%. 

#### 2.3.2. Rate of Acute Pyelonephritis

The comparison of the groups with antibiotic treatment (case) versus no treatment (control) on the rate of acute pyelonephritis is summarized in Figure 3. The analysis included six studies with 765 KT recipients, comparing antibiotics (310 KT recipients) versus no antibiotics for ASB (455 KT recipients). The studies by Fontsere and Antonio et al. showed statistical differences in favor of no antibiotic treatment with 7.4% versus 0.8% (*p* < 0.05), and 15% versus 10% (*p* < 0.08) rates of pyelonephritis, respectively [16,18]. The weight of these studies in the meta-analysis was, however, only 5.60% and 5.86%. The study by Coussement et al. represented 49.21% of the weight (OR 0.94, *p* = 0.87) [17]. Overall, there was no difference in the rate of pyelonephritis (OR 0.74, 95% CI 0.44 to 1.24, *p* = 0.25). The Q test for heterogeneity reported Q = 6.069, *p* = 0.300, tau2 = 0.000, H2 = 1.00 and I^2^ = 0%. 

#### 2.3.3. Hospitalization during Follow-Up

The comparison of groups with antibiotic treatment (case) versus no treatment (control) on hospitalization rate is summarized in Figure 4. The analysis included five studies with 590 KT recipients, comparing antibiotics (256 KT recipients) versus no antibiotics for ASB (334 KT recipients). The study by Antonio et al. showed a significantly higher hospitalization rate among patients receiving antibiotic treatment of 30% versus 4% of the control group [18]. However, the studies by Sabé and Coussement contributed the highest weights, of 29.98% and 21.82%, respectively [15,17]. Coussement measured hospitalization days and reported the mean hospitalization period (interquartile range) for both groups, which was 5 (3–36) days for those receiving antibiotics and 7 (5–13) for those without treatment [17]. Overall, there was no difference in days of hospitalization (OR 0.73, 95% CI 0.30 to 1.76, *p* = 0.48). The Q test for heterogeneity reported Q = 7.733, *p* = 0.102, tau2 = 0.485, H2 = 1.993, and I^2^ = 49.8%. 

#### 2.3.4. Serum Creatinine during Follow-Up

The comparison of the groups with antibiotic treatment (case) versus no treatment (control) on serum creatinine at the end of the follow-up is summarized in Figure 5. The analysis included three studies with 399 KT recipients, comparing antibiotics (165 KT recipients) versus no antibiotics for ASB (234 KT recipients). The study by Moradi et al. showed a statistically significant difference in serum creatinine, with higher values in the group receiving antibiotic treatment for ASB [10]. Overall, there was no difference in serum creatine between groups during follow-up (CI −0.41 to 0.12, *p* = 0.27). The Q test for homogeneity reported Q = 3.313 and *p* = 0.191.

#### 2.3.5. Rate of Graft Rejection during Follow-Up

The comparison of the groups with antibiotic treatment (case) versus no treatment (control) on the rate of graft rejection during follow-up is summarized in Figure 6. The analysis included four studies with 573 KT recipients, comparing antibiotics (165 KT recipients) versus no antibiotics for ASB (259 KT recipients). The study by Origen et al. represented 63.23% of the weight [13]. No study showed a statistical difference in graft rejection rate during follow-up. The meta-analysis showed OR 1.08, 95% CI 0.51 to 2.27 and *p* = 0.84. The Q test for heterogeneity reported Q = 0.485, *p* = 0.922, tau2 = 0.000, H2 = 1 and I^2^ = 0%. 

#### 2.3.6. Rate of Graft Loss during Follow-Up

The comparison of the groups with antibiotic treatment (case) versus no treatment (control) on the rate of graft loss during follow-up is summarized in Figure 7. The analysis included four studies with 598 KT recipients, comparing antibiotics (229 KT recipients) versus no antibiotics for ASB (369 KT recipients). The study by Coussement et al. represented 47.98% of the weight. No study showed a statistically significant difference in graft loss rate during follow-up [17]. The meta-analysis showed OR 1.31, 95% CI 0.37 to 4.59 and *p* = 0.67. The Q test for heterogeneity reported Q = 0.100, *p* = 0.992, tau2 = 0.000, H2 = 1 and I^2^ = 0%. 

#### 2.3.7. Rates of Multidrug-Resistant Microorganisms 

The comparison of groups with antibiotic treatment (case) versus no treatment (control) on the rate of multidrug-resistant microorganisms is summarized in Figure 8. The analysis included six studies with 632 KT recipients, comparing antibiotics (288 KT recipients) versus no antibiotics for ASB (344 KT recipients). The study by Antonino et al. reported a higher rate of MDR microorganisms in the group that did not receive antibiotics [18]. In the other studies, the rate of MDR microorganisms was higher in the group that did receive antibiotics, but without statistically significant differences. Overall, there was no difference in the rate of multidrug-resistant microorganisms between groups during follow-up (OR 0.65, 95% CI 0.29 to 1.46, *p* = 0.30). The Q test for heterogeneity reported Q = 12.235, *p* = 0.032, tau2 = 0.607, H2 = 2.583 and I^2^ = 61.3%.

#### 2.3.8. Mortality Rate 

The comparison of the group with antibiotic treatment (case) versus no antibiotics (control) on the mortality rate during the follow-up is summarized in Figure 9. The analysis included four studies with 503 KT recipients, comparing antibiotics (215 KT recipients) versus no treatment for ASB (288 KT recipients). The study by Coussement et al. represented 51.78% of the weight [17]. No study showed statistical difference in the mortality rate during follow-up. Overall, there was no difference in mortality rate during follow-up (logOR −0.57, 95% CI −1.66 to 0.53, *p* = 0.31). The Q test for heterogeneity reported Q = 0.354, *p* = 0.950, tau2 = 0.000, H2 = 1.000 and I^2^ = 0%.

#### 2.3.9. Risk of Bias Assessment 

The selection and classification of patients did not show bias that could influence the results in the randomized, comparative and prospective studies. However, in other studies, treatment decisions were based on the physician’s evaluation in each case, and this might have affected the results. Three studies had a serious risk of bias regarding the selection of study participants. The risks of bias of the included studies are summarized in Figure 10).

### 2.4. Evidence Synthesis

The recommendations to test for and treat asymptomatic bacteriuria after kidney transplantation have traditionally been based on the studies by Ramsey et al. and Prát et al., who found rates of ASB of 91% and 96%, and rates of urinary tract infections after KT of 54% and 62%, respectively [21,22]. Both studies were observational and retrospective and did not focus on the effect of the antibiotic treatment of ASB on the incidence of urinary tract infections, pyelonephritis and graft outcomes. The acquired evidence in our systematic review demonstrates, however, that screening and antibiotic treatment of asymptomatic bacteriuria does not reduce the rates of acute urinary tract infections, pyelonephritis, graft rejection, graft loss isolation of multidrug-resistant microorganisms or all-cause mortality within the first year after KT. The findings also mean that it was not possible to identify patient subgroups with specific risk factors that would benefit from the antibiotic treatment of ASB [13,15,23]. 

Different regimens of antibiotics were used in the studies included. The type of antibiotic was selected in many cases by the physician. The duration of the treatment was predefined in the comparative studies: 10 days in the article by Coussement et al. and Moradi et al. [10,17] and 3 to 7 days in most of the comparative and observational studies included in the analysis [12,13,14,15,18]. The research by Coussement et al. reported that, at the beginning of the study, fluoroquinolones were the most commonly prescribed antibiotics, followed by second-/third-generation cephalosporins, amoxicillin and amoxicillin–clavulanic acid, and the microbiological analysis reported the isolation of Enterobacteriaceae in 87% in the study [17]. *E. coli* and *Klebsiella* spp. represent 43–64% and 15–17%, respectively. Moreover, *Pseudomonas* spp. was isolated in 2–7.9% of the positive cultures [12,13,14,15,16,17,18]. The research by Sabé et al. reported a higher percentage of *Klebsiella* spp. isolation, during follow-up, in the group receiving antibiotics (31.1% versus 17.5%) and a lower percentage of *E. coli* (36% versus 54%) [15]. The studies by Coussement [17] and Antonio [18] showed contradictory findings on the effect of the antibiotic treatment of ASB on the rate of isolation of MDR organisms, which, most likely, is due to different study characteristics. Coussement found a higher rate of MDR organisms in the intervention group, while Antonio found a higher rate in the control group. The study by Coussement et al. [17] included patients beyond two months after KT with at least one episode of asymptomatic bacteriuria. Investigators reported a higher incidence of asymptomatic bacteriuria in the group with no treatment, compared to patients that received antibiotics from the first month of follow-up, with incidences of 66% and 29%, respectively. This difference was confirmed at the end of follow-up, with incidences of 53% and 33%. Antibiotics were administered for 10 days to patients in the treatment group. The duration of treatment might explain the higher resistance rate in this group. Moreover, the mean antibiotic consumption was five times higher among patients in the treatment group. Antonio et al. evaluated the incidence of asymptomatic bacteriuria and the cumulative incidence of infection in the two first months after kidney transplantation. The mean time with the ureteral catheter was 64 days, which means that a high percentage of patients had ureteral catheters during the study period. The isolation of ESBL-producing *E. coli* was also higher in the group with no treatment (22.5% vs. 7.5%). Finally, the use of antibiotics for other reasons was 12.8% in the control group and 17.5% in the group receiving antibiotics (*p* = 0.56). The duration of antibiotic treatment was 5 days in the treatment group. This may explain the lower incidence of MDR microorganisms related to less antibiotic treatment [13,15,20].

## 3. Discussion

ASB is highly prevalent after KT, especially in the first six months after transplantation [7]. The indications for diagnosis and treatment of ASB have mainly been to prevent pyelonephritis in recipients with risk factors such as stents, catheters, neurogenic bladder, female gender, glomerulonephritis as the cause of end-stage renal disease, double renal transplant or combined transplants [5,23]. On this background it is surprising that even carefully designed randomized controlled trials do not show evidence for a benefit of treatment of ASB. 

The diagnosis and treatment of ASB has been extensively studied for many years in pregnant women and before urological interventions, and there are still controversies related to clinical management [24]. We believe that the lack of evidence, related to the management of ASB in all these fields, is due to confounding risk factors that are not being considered in the stratification of study patients. No other group at risk of developing UTI has a larger variety of or more important risk factors than KT recipients. Among KT recipients, stratification for patients with urinary catheters, those with incomplete bladder emptying or recurrent urinary infections may give some evidence about the risk of ASB, symptomatic UTIs, and the potential effect of the antibiotic treatment. A successful KT may add up to 20 years of life in young patients. According to the O`Neill report, antimicrobial resistance will increase in the coming decades, and this development is already ahead of schedule [25,26]. KT recipients are, therefore, not only at risk of developing UTI, but of developing pan-resistant infections. Twenty thousand KTs are performed in Europe every year [26]. This means that the population of KT patients is several hundred thousand in Europe only.

The strengths of the present review are the strict inclusion criteria and the identification of randomized controlled studies. The main weaknesses are the lack of significant findings in individual studies and the contradictory findings among studies in the meta-analysis. The analysis of comparative non-randomized studies and observational studies did not provide supplementary evidence to guide clinical recommendations. We believe that the lack of significant findings is due to confounding risk factors that are not being considered in the stratification of study patients. We also believe that these factors were not identified in the homogeneity analyses that were performed as part of our meta-analyses. Our focus on ASB during the first year after KT limited the number of available records.

Our findings demonstrate an unacceptable lack of evidence related to the diagnosis and treatment of ASB within the first year after kidney transplantation. This knowledge gap might put a large patient population at risk of developing infections with pan-resistant micro-organisms, and better studies are therefore urgently needed. To overcome the limitations of poor patient stratification in previous studies, we need larger studies, which preferably should be embedded in a registry. This will enable us to evaluate groups of patients with risk factors such as diabetes mellitus, neurogenic bladder, recurrent urinary tract infections, anatomical abnormalities and urinary tract catheters. To identify the most important risk factors, the protocol development should preferably be preceded by a multidisciplinary consensus process. Specific knowledge gaps that need to be filled include whether the management of asymptomatic bacteriuria after kidney transplantation is required in the early post-operative period. Moreover, we must study if asymptomatic bacteriuria requires treatment in patients with urinary catheters and if antibiotic prophylaxis is needed before the removal of bladder and ureteral catheters. A key objective of future studies must be to determine if the treatment of asymptomatic bacteriuria increases antibiotic resistance among uropathogens.

## 4. Materials and Methods

### 4.1. Definitions Used 

According to our inclusion criteria, symptomatic bacteriuria (ASB) was defined as the isolation of a single bacterial species with >10^5^ CFU/in two urine specimens from a patient without symptoms of UTI [9]. However, most studies only used one specimen for diagnosis of ASB, and, therefore, both definitions were accepted for inclusion in our analysis. Urinary tract infection (UTI) was defined as the clinical symptoms of UTI in a patient with monomicrobial growth, due to invasion of the urinary tract, and comprised both lower UTI and acute pyelonephritis. Lower UTI was defined by irritative voiding symptoms (dysuria, frequency or urgency) and the presence of bacteriuria in the absence of diagnostic criteria for pyelonephritis. Acute pyelonephritis was defined by the simultaneous presence of fever and bacteriuria and/or bloodstream infection, along with at least one of the following: flank pain, graft pain, chills and/or irritative voiding symptoms [27].

Graft loss was defined as loss of kidney function necessitating chronic dialysis [3]. The diagnosis of allograft rejection was based on impaired renal function and findings on renal biopsy. The Banff classification was used for evaluation and diagnosis of graft rejection [3,28,29]. Hospitalization was defined as any hospital admission, for whatever reason both as outpatients and in-patients. Mortality was defined as death due to any cause during study follow-up.

### 4.2. Study Variables and Outcomes

The primary outcome was the effect of screening and antibiotic treatment of asymptomatic bacteriuria after kidney transplantation on the incidence of symptomatic urinary tract infections. The secondary outcome variables were as follows: incidence of acute pyelonephritis; graft function during follow-up; rate of acute graft rejection; severity of urinary tract infections; rate of hospitalization; mortality rate; microbiological profile; and rates of multidrug-resistant (MDR) microorganisms in kidney transplant recipients with and without antibiotic treatment of asymptomatic bacteriuria and urinary tract infections. The systematic review was preregistered at PROSPERO (International Prospective Register of Systematic Reviews), and the detailed review protocol can be viewed under the Centre for Reviews and Dissemination (CRD) number 42024511920. This is available from the following link: https://www.crd.york.ac.uk/prospero/display_record.php?ID=CRD42024511920 (accessed on 1 May 2024)

### 4.3. Literature Search

A systematic review was conducted according to the Preferred Reporting Items for Systematic Reviews and Meta-Analyses (PRISMA) guidelines [19]. The following PICO questions were formulated: Patients (adult patients receiving a kidney transplantation), Intervention (screening and treatment of ASB), Control (screening and no antibiotic treatment of ASB) and Outcome (rates of urinary tract infections, sepsis and kidney failure). 

The search was made in MEDLINE (Pubmed), the Cochrane Library CENTRAL and Embase in the period from January 1979 to January 2024. The search terms used were “asymptomatic bacteriuria” and “kidney transplantation” or “kidney transplant”. Inclusion criteria were manuscripts in English addressing the management of asymptomatic bacteriuria and kidney transplantation in adult males and females (>16 years), which compared the effect of diagnosis and treatment of ASB with no diagnosis and treatment of ASB (intervention and non-intervention). Hence, the intervention was defined as screening and antibiotic treatment for asymptomatic bacteriuria in kidney transplant recipients. The control group was defined as kidney transplant recipients who were screened and did not receive antibiotic treatment for asymptomatic bacteriuria. 

### 4.4. Eligibility of Studies

This review identified randomized controlled trials (RCTs), as well as prospective clinical trials (non-RCTs) and retrospective cohort studies, cross-sectional studies, case–control studies and single-arm studies (with at least 10 patients). If more than one publication evaluated the same patient cohort, only the larger and more comprehensive publication was included. Case reports, expert opinions, comments, editorials and conference abstracts were excluded from review. Studies that included patients with urinary catheters, such as urethral catheters and double J stents, were also not included in the analysis, as they have a higher risk of infections (Figure 1). Only studies that documented the effect of antibiotic treatment versus no treatment of ASB in KT recipients were included in the meta-analysis. 

### 4.5. Selection of Studies and Data Extraction 

Studies were selected in a two-step procedure. In the first step, the titles and abstracts were screened by applying the inclusion criteria. In the second step, the full text was examined, again according to the inclusion criteria. Two authors undertook the study selection independently. Any disagreements were resolved through discussion with a third author. Data extraction was also performed independently by two authors, using a preset Excel sheet containing all study variables as registered at PROSPERO.

### 4.6. Risk of Bias Assessment

Two authors independently assessed the risk of bias with the Cochrane Risk-of-Bias tool [30]. The following domains of bias were included: selection bias, performance bias, classifications of interventions, missing data, measurement of the outcomes and reporting bias. Any disagreements were resolved by the involvement of a third reviewer. For non-randomized trials, the ROBINS-I-tool Version 2016 (risk of bias in non-randomized studies of interventions) was used [30].

### 4.7. Meta-Analysis

Meta-analysis was performed using a random effects model. Inter-study heterogeneity was assessed visually using forest plots and the I² statistics with 95% CI and Chi-squared p-values for heterogeneity. The analysis included the estimated average log odds ratio based on the random effects model and the Q test for heterogeneity. Publication bias was assessed using Egger’s test and by visual inspection of the funnel plot. The meta-analysis was performed using Statistical Package for the Social Sciences (SPSS) Statistics for Macintosh (MAC), version 29.0.2.0 (International Business Machines (IBM) Corporation, Armonk, NY, USA). The meta-analysis analyzed the influence of treatment of ASB following KT on key study variables.

## 5. Conclusions

There is no evidence that the antibiotic treatment of asymptomatic bacteriuria is associated with a lower incidence of UTIs and acute pyelonephritis, with a positive effect on graft survival rate in adult KT recipients within the first year after transplantation. The absence of evidence for the benefits of the antibiotic treatment of ASB means that there is also no evidence for the risk of unwanted effects from not treating ASB in the first year after KT. 

## Figures and Tables

**Figure 1 antibiotics-13-00442-f001:**
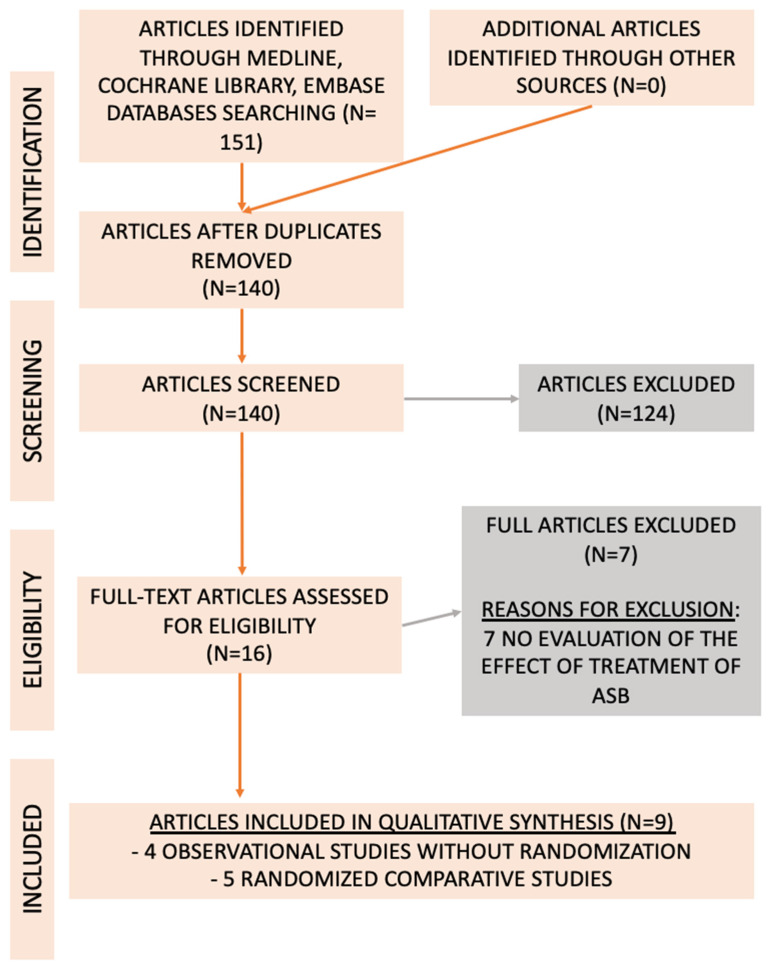
Flowchart with the number of publications evaluated and included in the analysis, according to the Preferred Reporting Items for Systematic Reviews and Meta-Analyses (PRISMA) guidelines [19].

**Figure 2 antibiotics-13-00442-f002:**
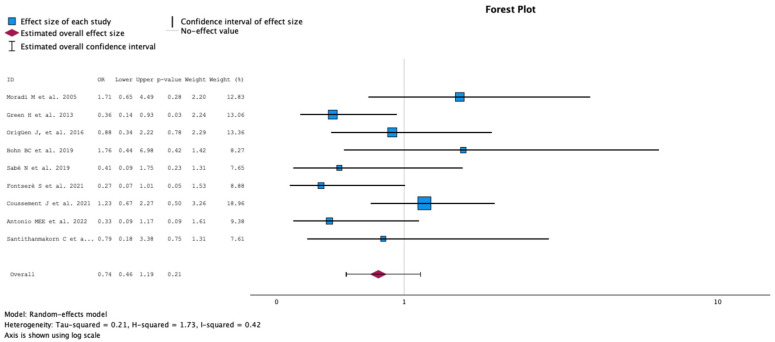
Analysis of the comparison of the no-treatment groups versus the antibiotics groups in the outcome rate of symptomatic urinary tract infections [10,12,13,14,15,16,17,18,20].

**Figure 3 antibiotics-13-00442-f003:**
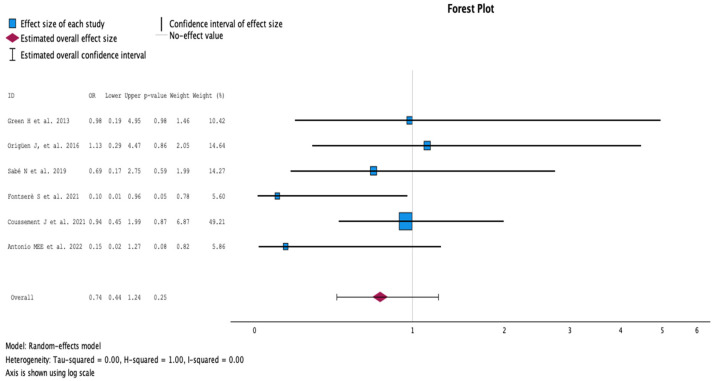
Analysis of the comparison of the no-treatment groups versus the antibiotics groups in the outcome rate of acute pyelonephritis [12,13,15,16,17,18].

**Figure 4 antibiotics-13-00442-f004:**
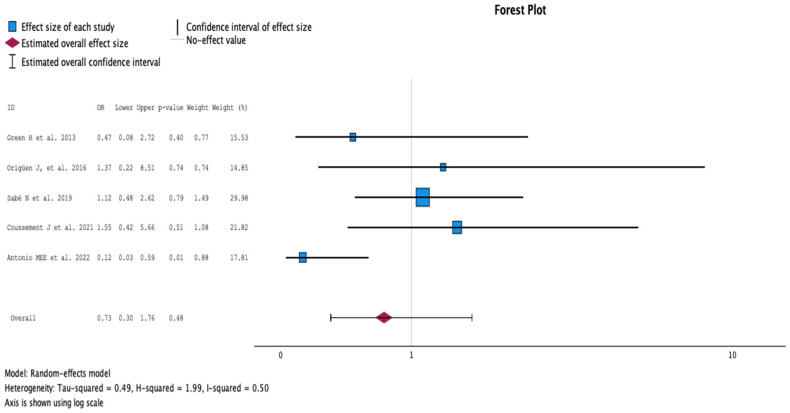
Analysis of the comparison of the no-treatment groups versus the antibiotics groups in the outcome hospitalization rate [12,13,15,16,18].

**Figure 5 antibiotics-13-00442-f005:**
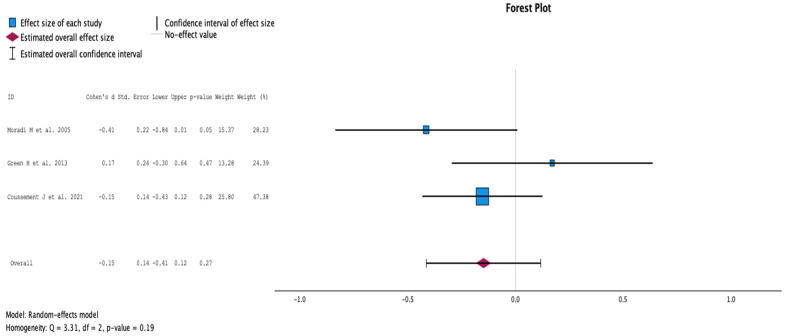
Analysis of the comparison of the no-treatment groups versus the antibiotics groups in the outcome serum creatinine at the end of the follow-up [10,12,18].

**Figure 6 antibiotics-13-00442-f006:**
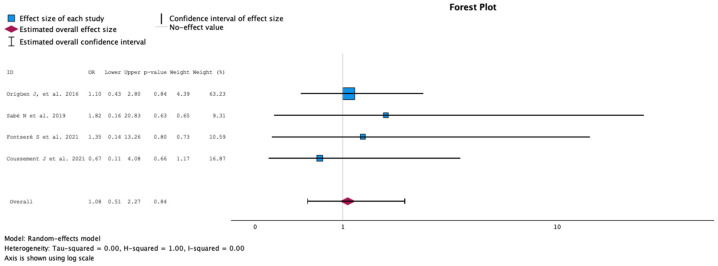
Analysis of the comparison of the no-treatment groups versus the antibiotics groups in the outcome rate of graft rejection during the follow-up [13,15,16,17].

**Figure 7 antibiotics-13-00442-f007:**
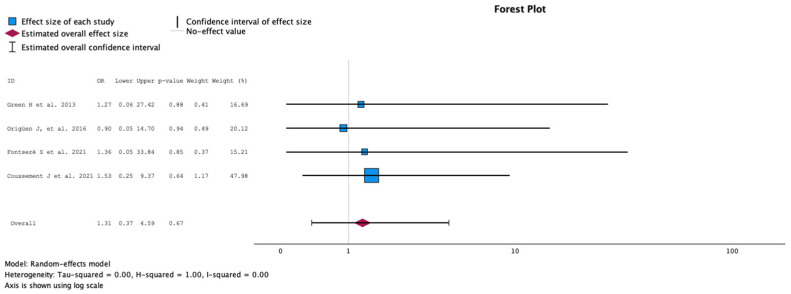
Analysis of the comparison of the no-treatment groups versus the antibiotics groups in the outcome rate of graft loss during the follow-up [12,13,16,17].

**Figure 8 antibiotics-13-00442-f008:**
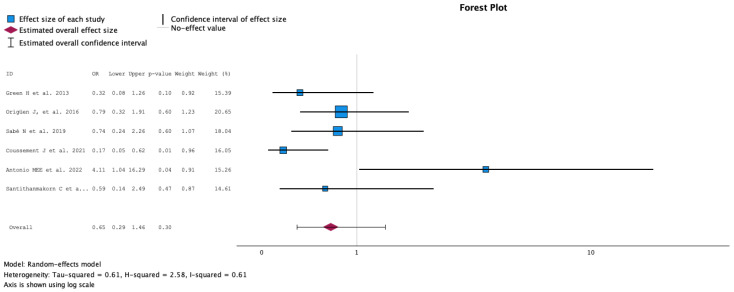
Analysis of the comparison of the no-treatment groups versus the antibiotics groups in the outcome rate of multidrug-resistant microorganisms [12,13,15,17,18,20].

**Figure 9 antibiotics-13-00442-f009:**
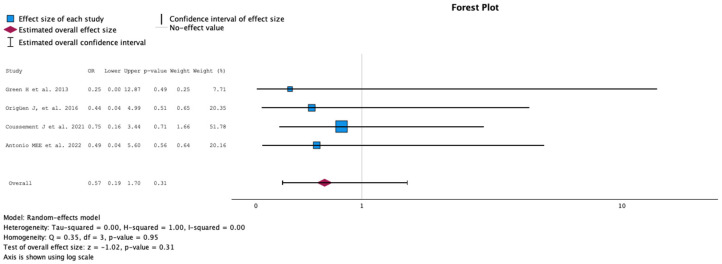
Analysis of the comparison of the no-treatment groups versus the antibiotics groups in the outcome mortality rate during the follow-up [12,13,17,18].

**Figure 10 antibiotics-13-00442-f010:**
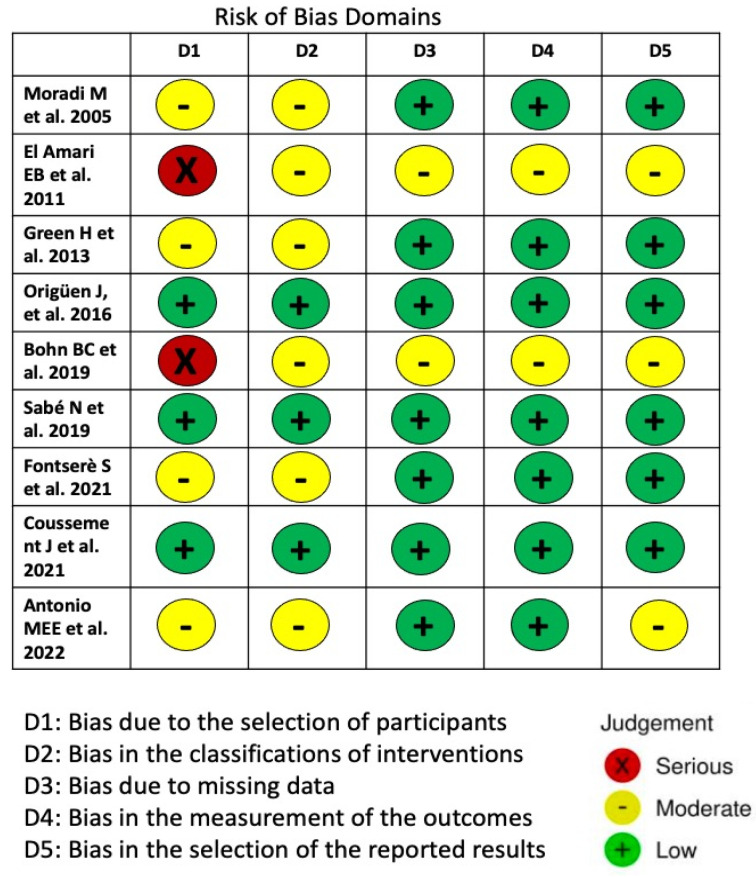
Risk of bias assessment for the nine included randomized, comparative and observational studies utilizing the risk of bias in non-randomized studies of interventions (ROBINS-I) tool [10,11,12,13,14,15,16,17,18].

**Table 1 antibiotics-13-00442-t001:** Characteristics of randomized studies.

	Coussement, J. et al., 2021 [17]	Origüen, J. et al., 2016 [13]	Moradi, M. et al., 2005 [10]	Sabé, N. et al., 2019 [15]	Antonio, M.E.E. et al., 2022 [18]
Type of study	Prospective and randomized	Prospective and randomized	Prospective and randomized	Prospective and randomized	Randomized, observational and controlled
Primary outcome	Rate of symptomatic UTIS	Rate of acute pyelonephritis	Rate of ASB and symptomatic UTIs	Rate of acute pyelonephritis	Rate of and time to UTI and acute pyelonephritis
Intervention strategy	Antibiotic treatment for 10 days	Antibiotic treatment	Antibiotic treatment for 10 days	Antibiotic treatment for 5–7 days	Antibiotic treatment
Control strategy	No treatment	No treatment	No treatment	No treatment	No treatment
Inclusion criteria	ASB 2 months after KT (screening of ASB required)	ASB 2 months after KT (screening of ASB required)	ASB one year after KT (screening of ASB required)	KT after urinary catheter removed (ureteral and bladder catheter)	KT after removal of the urethral catheter
Follow-up period after inclusion	12 months	24 months	12 months	12 months	2 months
Number of patients randomized	199 (100 cases and 99 controls)	112 (53 cases and 59 controls)	88 (43 cases and 45 controls)	87 (41 cases and 46 controls)	80 (40 cases and 40 controls)

Definition of asymptomatic bacteriuria (ASB): 10^5^ CFU/mL; KT: kidney transplantation; UTIs: urinary tract infections; ASB: asymptomatic bacteriuria.

**Table 2 antibiotics-13-00442-t002:** Findings in randomized studies.

		Coussement, J. et al., 2021 [17]		Origüen, J. et al., 2016 [13]		Moradi, M. et al., 2005 [10]		Sabé, N. et al., 2019 [15]		Antonio, M.E.E. et al., 2022 [18]	
		Intervention Group	Control Group	Intervention Group	Control Group	Intervention Group	Control Group	Intervention Group	Control Group	Intervention Group	Control Group
Infection variables	Rate of UTI	27% (27/100)	31% (31/99) (*p* = 0.49)	20.7% (11/53)	18.6% (11/59) (*p* = 0.78)	21% (9/43)	31% (14/31)	14.6% (6/41)	6.5% (3/46) (*p* = 0.215)	25% (10/40)	10% (4/40) (*p* = 0.07)
	Rate of PNA	17% (17/100)	16% (16/99) (*p* = 0.87)	7.5% (4/53)	8.4% (5/59) (*p* = 1)	ND	ND	12.2% (5/41)	8.7% (4/46) (*p* = 0.59)	15% (6/40)	10% (1/2.5) (*p* = 0.04)
	Rate of ASB	29% (27/92)	66% (62/94) (*p* < 0.001)			58.1% (25/43)	73.3% (33/45)	42.3% (41/102)	50.5% (46/103)	17.5% (7/40)	37.5% (15/40) (*p* = 0.045)
Kidney transplant variables	Rejection	3% (3/100)	2% (2/99) (*p* = 1)	18.9% (10/53)	20.3% (12/59) (*p* = 0.84)	ND	ND	2.4% (1/41)	4.3% (2/46) (*p* = 0.63)	ND	ND
	Graft loss	2% (2/100)	3% (3/100) (*p* = 0.68)	1.9% (1/53)	1.7% (1/59) (*p* = 1)	ND	ND	ND	ND	ND	ND
General variables	Hospitalization	4% (4/100)	6% (6/99) (*p* = 0.51)	3.7% (2/53)	5.1% (3/59) (*p* = 0.73)	ND	ND	53.7% (22/41)	56.5% (26/46) (*p* = 0.83)	30% (12/40)	5% (2/40) (*p* < 0.01)
	Mortality	4% (4/100)	3% (3/99) (*p* = 1)	3.8% (2/53)	1.7% (1/59) (*p* = 0.60)	ND	ND	ND	ND	5% (2/40)	2.5% (1/40) (*p* = 1)
Microbiological variables	Rate of MDR	18% (13/72)	4% (3/83) (*p* = 0.003)	24.5% (13/53)	20.3% (12/59) (*p* = 0.65)	ND	ND	19.5% (8 ESBL /41)	15.2% (7 ESBL/46)	ESBL *E. coli* and *Klebsiella* spp. 12.5% (3/40)	ESBL *E. coli* and *Klebsiella* spp. 25% (10/40)
	Rate of *E. coli* as pathogen	70% (19/27)	61% (19/31)	51.5% (105/204)	36.2% (85/235)	69.7%	60%	36.1% (43/119)	54% (74/137)	25% (10/40)	30% (12/40) (*p* = 0.61)
	Rate of *Klebsiella* spp. as pathogen	4% (1/27)	13% (4/31)	17.1% (39/204)	27.0% (61/235)	6.9%	13.4%	31.1% (37/119)	17.5% (24/137)	25% (10/40)	7.5% (3/40) (*p* = 1)
	Rate of *Pseudomonas* spp. as pathogen	ND	ND	4.4% (9/204)	9.8% (23/235)	9.3%	6.7%	6.7% (8/119)	5.8% (8/137)	ND	ND

Abbreviations used: asymptomatic bacteriuria (ASB) definition (10^5^CFU/mL); KT: kidney transplantation; UTIs: urinary tract infections; PNA: acute pyelonephritis; MDR: multidrug-resistant microorganisms; ESBL: extended-spectrum beta-lactamase-producing bacteria; ND: no data described.

**Table 3 antibiotics-13-00442-t003:** Characteristics of observational studies.

	Green, H. et al., 2013 [12]	Bohn, B.C. et al., 2019 [14]	Fontserè, S. et al., 2021 [16]	Santithanmakorn et al., 2022 [20]
Type of study	Single centre, retrospective and observational	Single centre, observational and retrospective	Single centre and prospective	Single centre and retrospective study
Inclusion criteria	KT performed in a centre with a minimum follow-up of 6–12 months	KT beyond 1 year follow-up and positive screening for ASB	KT with positive screening for ASB or diagnosis of UTIs during follow-up	KT performed in the centre
Primary outcome	Rate of hospitalization due to UTIs and reduction eGFR	Rate of progression from ASB to UTI	Cases of ASB and symptomatic UTIs; effect of treatment of ASB was a secondary objective	Incidence of UTIs after KT (evaluation of treatment of ASB was secondary objective)
Intervention strategy	Antibiotic treatment for ASB (decision to treat ASB was based on physician decision in each case)	Antibiotic treatment for ASB	Antibiotic treatment for ASB	Antibiotic treatment for ASB
Control strategy	No treatment for ASB	No treatment for ASB	No treatment for ASB	No treatment for ASB
Follow-up period after inclusion	12 months	12 months	6 months	12 months
Number of patients	112 (22 cases and 90 controls)	64 (53 cases and 11 controls)	175 (54 cases and 121 controls)	42 (32 cases and 10 controls)

Def of ASB (10^5^ CFU/ml); KT: kidney transplantation; UTIs: urinary tract infections; ASB: asymptomatic bacteriuria, eGFR: estimated glomerular filtration rate.

**Table 4 antibiotics-13-00442-t004:** Findings in observational studies.

		Green, H. et al., 2013 [12]		Bohn, B.C. et al., 2019 [14]		Fontserè, S. et al., 2021 [16]		Santithanmakorn et al., 2022 [20]	
		Intervention Group	Control Group	Intervention Group	Control Group	Intervention Group	Control Group	Intervention Group	Control Group
Infection variables	Rate of UTI	54% (12/22)	30% (27/90) (*p* < 0.05)	11.1% (6/54)	2.5% (4/121) (*p* = 0.007)	25% (13/53)	36% (4/11) (*p* = 0.463)	65.6% (21/32)	60% (6/10)
	Rate of PNA	9.1% (2/22)	8.9% (8/90)	7.4% (4/54)	0.8% (1/121) (*p* = 0.003)	ND	ND	ND	ND
	Rate of ASB	40% (12/22)	54% (36/90)	44.4% (24/54)	40.5% (49/121) (*p* = 0.8)	ND	ND	ND	ND
Kidney transplant variables	Rejection	ND	ND	1.8% (1/54)	2.5% (3/121) (*p* = 0.8)	ND	ND	ND	ND
	Graft loss	0%	2.2% (2/90)	0% (0/54)	0.8%(1/121) (*p* = 0.7)	ND	ND	ND	ND
General variables	Hospitalization	9.1%	4.4% (*p* < 0.026)	ND	ND	ND	ND	ND	ND
	Mortality	0%	0%	ND	ND	ND	ND	ND	ND
Microbiological variables	Rate of MDR	36%	ND	ND	ND	ND	ND	53.1% (17/32)	40% (4/10) (*p* < 0.001)
	*E. coli*	N = 13	N = 30	ND	ND	ND	ND	ND	ND
	*Klebsiella* spp.	N = 6	N = 21	ND	ND	ND	ND	ND	ND
	*Pseudomonas* spp.	N = 0	N = 8	ND	ND	ND	ND	ND	ND

Definition of asymptomatic bacteriuria (ASB) (10^5^CFU/mL); KT: kidney transplantation; UTIs: urinary tract infections; ASB: asymptomatic bacteriuria; PNA: acute pyelonephritis; MDR: multidrug-resistant microorganisms; ND: no data described.

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
