# Peer review of "Systematic Review and Meta-Analysis Provide no Guidance on Management of Asymptomatic Bacteriuria within the First Year after Kidney Transplantation"

_antibiotics, 2024, doi:10.3390/antibiotics13050442_

Round 1
Reviewer 1 Report
Comments and Suggestions for Authors
The authors have conducted a systematic review with meta-analysis to review the current evidence and knowledge gaps related to screening and treatment of asymptomatic bacteriuria during specific time periods after kidney transplantation. Life expectancy has increased in recent years due to medical advances. One of them is the possibility of performing organ transplants. The authors have investigated a relevant issue such as kidney rejection in the face of a urinary infection in these patients. The authors carried out a systematic review to try to increase knowledge about asymptomatic bacteriuria in kidney transplant recipients and medical conduct regarding whether or not antimicrobial treatment is necessary. A systematic review with meta-analysis presents the highest level of scientific evidence because it compares works published under strict quality guidelines. The results may contribute to medical decision making in patients with symptomatic bacteriuria who will receive a kidney transplant.
The methodology is correct and complete. As the authors report, the main limitation is the small number of randomized and randomized publications. The authors describe the different aspects to be discussed about the results and present statistical evidence that accompanies the discussion and conclusions. References are appropriate. Most of the appointments are current and in quantity according to the work. The figures are appropriate and in quantity according to the work. The tables need corrections such as the addition of abbreviations and italics in the names of bacteria (E. coli, Klebsiella spp. and Pseudomonas spp.).
Observations (see archive)
-the objective of the study is worded differently in the abstract and in the introduction
-abbreviations need to be clarified and there is a paragraph that begins with a number (124) in the introduction
-italics are missing in bacterial names
-there are some misspelled words (Ttable, et all)
-et al or et al.? Unify formatting throughout the manuscript
-there are some paragraphs with different font color
-reference 29 needs correction.
Author Response
COMMENTS TO REVIEWER
Reviewer: 1
The authors have conducted a systematic review with meta-analysis to review the current evidence and knowledge gaps related to screening and treatment of asymptomatic bacteriuria during specific time periods after kidney transplantation. Life expectancy has increased in recent years due to medical advances. One of them is the possibility of performing organ transplants. The authors have investigated a relevant issue such as kidney rejection in the face of a urinary infection in these patients. The authors carried out a systematic review to try to increase knowledge about asymptomatic bacteriuria in kidney transplant recipients and medical conduct regarding whether or not antimicrobial treatment is necessary. A systematic review with meta-analysis presents the highest level of scientific evidence because it compares works published under strict quality guidelines. The results may contribute to medical decision making in patients with symptomatic bacteriuria who will receive a kidney transplant.
The methodology is correct and complete. As the authors report, the main limitation is the small number of randomized and randomized publications. The authors describe the different aspects to be discussed about the results and present statistical evidence that accompanies the discussion and conclusions. References are appropriate. Most of the appointments are current and in quantity according to the work. The figures are appropriate and in quantity according to the work. The tables need corrections such as the addition of abbreviations and italics in the names of bacteria (E. coli, Klebsiella spp. and Pseudomonas spp.).
We thank the reviewer for these comments. The naming of the microorganisms are edited to italics according to nomenclature recommendations
Observations (see archive)
-the objective of the study is worded differently in the abstract and in the introduction
The objective has been corrected and is now similar in the abstract (adapted to word limit) and the introduction:
Abstract: We aimed to evaluate available evidence regarding the benefit of screening and treatment of ASB within the first year after KT.
Introduction: The aim of our systematic review and meta-analyses was to review current evidence to try and fill identified knowledge gaps related to a possible benefit screening and treatment of ASB during specific time periods after KT (two, six or twelve months) as well as in recipients with specific risk factors for UTI.
-abbreviations need to be clarified and there is a paragraph that begins with a number (124) in the introduction
Corrected
-italics are missing in bacterial names
Corrected
-there are some misspelled words (Ttable, et all)
Corrected
et al. was edited to italics
-et al or et al.? Unify formatting throughout the manuscript
et al. was unified to et al. (in italics) throughout the manuscript
-there are some paragraphs with different font color
Thank you, we have corrected all headings to black (in the last part of the discussion)
-reference 29 needs correction
This reference has been edited in the text and in references
Reviewer 2 Report
Comments and Suggestions for Authors
The work entitled “SYSTEMATIC REVIEW AND META-ANALYSIS PROVIDE NO GUIDANCE ON MANAGEMENT OF ASYMPTOMATIC BACTERIURIA WITHIN THE FIRST YEAR AFTER KIDNEY TRANSPLANTATION” provides an analysis of available studies/data on the utility of antibiotic therapy and related outcome upon ABS detection. The paper is well written, clearly and understandable also by a non professional audience. Nevertheless, some acronyms should be clarified for everybody that is not familiar with the medical jargon:
Please define in lines 63-64 the acronyms EAU ans IDSA. Please so the same with any acronym before its introduction.
Define PROSPERO at line 364.
Define CRD at line 365.
Minor corrections:
As assessed by Digital Color Meter the text color in lines150-151 is not black, please modify for homogeneity. Same in lines 165-166, 181-182, etc. Please harmonize the whole document for the text to be in black.
Final remarks:
Although the paper disclose a lack or correlation between antibiotic treatment and patient outcome upon ASB, for the general reader of antibiotics it would be interesting the detail of which antibiotic or antibiotics are used (and at which concentration) and also which kind of bacteria are commonly found in ASB. Please, elaborate a little bit on these details.
Importantly, although you conclude that “lack of sig
nificant findings is due to confounding risk factors that are not being considered in the
stratification of study patients...”. Optionally, could you elaborate more on what could be those “confounding risk factors” (just an educated guess)?
If all the corrections and suggestions are implemented, the paper should be published without further round of revisions.
My best,
The reviewer.
Author Response
Reviewer: 2
The work entitled “SYSTEMATIC REVIEW AND META-ANALYSIS PROVIDE NO GUIDANCE ON MANAGEMENT OF ASYMPTOMATIC BACTERIURIA WITHIN THE FIRST YEAR AFTER KIDNEY TRANSPLANTATION” provides an analysis of available studies/data on the utility of antibiotic therapy and related outcome upon ABS detection. The paper is well written, clearly and understandable also by a non professional audience. Nevertheless, some acronyms should be clarified for everybody that is not familiar with the medical jargon:
Please define in lines 63-64 the acronyms EAU ans IDSA. Please so the same with any acronym before its introduction.
We thank the reviewer for these comments
Acronyms have been revised and clarified.
Define PROSPERO at line 364.
PROSPERO (International Prospective Register of Systematic Reviews)
Define CRD at line 365.
Centre for Reviews and Dissemination (CRD) number 42024511920. Available from: https://www.crd.york.ac.uk/prospero/display_record.php?ID=CRD42024511920
Minor corrections:
As assessed by Digital Color Meter the text color in lines150-151 is not black, please modify for homogeneity. Same in lines 165-166, 181-182, etc. Please harmonize the whole document for the text to be in black.
We have corrected and harmonized the whole document to black
Final remarks:
Although the paper disclose a lack or correlation between antibiotic treatment and patient outcome upon ASB, for the general reader of antibiotics it would be interesting the detail of which antibiotic or antibiotics are used (and at which concentration) and also which kind of bacteria are commonly found in ASB. Please, elaborate a little bit on these details.
Different regimens of antibiotics were used in the studies included. The type of antibiotic was selected in many cases by the physician. The duration of the treatment was predefined in the comparative studies: 10 days in the article by Coussement et al. and Moradi et al. [15, 22], 3 to 7 days in most of the comparative and observational studies included in the analysis [17, 18, 20, 23, 24]. The research by Coussement et al. reported that at the beginning of the study, fluoroquinolones were the most commonly prescribed antibiotics, followed by second-/third-generation cephalosporins, amoxicillin and amoxicillin-clavulanic acid and the microbiological analysis reported isolation of Enterobacteriaceae in 87% in the study [22]. E. coli and Klebsiella spp. represent 43-64% and 15-17%, respectively. Moreover, Pseudomonas spp. was isolated in 2-7.9% of the positive cultures [15, 17, 18, 23, 24]. The research by Sabé et al. reported a higher percentage of Klebsiella spp. isolation, during follow-up, in the group receiving antibiotics (31.1% versus 17.5%) and a lower percentage of E. coli (36% versus 54%) [20].
Importantly, although you conclude that “l ack of significant findings is due to confounding risk factors that are not being considered in the stratification of study patients...”. Optionally, could you elaborate more on what could be those “confounding risk factors” (just an educated guess)?
We believe lack of evidence related to management of ASB in all these fields is due to confounding risk factors that are not being considered in the stratification of study patients. No other group at risk of developing UTI has a larger variety, and more important risk factors than KT recipients. Among KT recipients stratification for patients with urinary catheters, those with incomplete bladder empty or recurrent urinary infections may give some evidence about the risk of ASB, symptomatic UTIs and the potential effect of the antibiotic treatment.
If all the corrections and suggestions are implemented, the paper should be published without further round of revisions.
My best,
The reviewer.